# Aromatic Characteristics of Passion Fruit Wines Measured by E-Nose, GC-Quadrupole MS, GC-Orbitrap-MS and Sensory Evaluation

**DOI:** 10.3390/foods11233789

**Published:** 2022-11-24

**Authors:** Ruojin Liu, Yaran Liu, Yuxuan Zhu, Maaria Kortesniemi, Baoqing Zhu, Hehe Li

**Affiliations:** 1Beijing Key Laboratory of Food Processing and Safety, Department of Food Science, College of Biological Sciences and Biotechnology, Beijing Forestry University, Beijing 100083, China; 2Food Sciences, Department of Life Technologies, University of Turku, FI-20014 Turku, Finland; 3Key Laboratory of Brewing Molecular Engineering of China Light Industry, Beijing Technology and Business University, Beijing 100048, China

**Keywords:** passion fruit wine, volatiles, sensory attributes, aroma, *Saccharomyces cerevisiae*

## Abstract

This study investigated the volatile composition and aromatic features of passion fruit wines using a combination of gas chromatography–quadrupole mass spectrometry (GC-qMS), gas chromatography–Orbitrap–mass spectrometry (GC-Orbitrap-MS), electronic nose (E-nose) and sensory evaluation. The results showed that these passion fruit wines possessed different aromatic features confirmed by E-nose. Seventeen sulfur compounds and seventy-eight volatiles were detected in these passion fruit wines using GC-Orbitrap-MS and GC-qMS, respectively. Forty-four volatiles significantly contributed to the overall wine aroma. These wines possessed passion fruit, mango, green apple, lemon and floral aromas confirmed by sensory evaluation. The partial least squares regression analysis indicated that sulfides, esters and terpenes, and terpenes mainly correlated to the passion fruit, mango and green apple aroma, respectively. Sulfur compounds significantly affected the aroma of passion fruit wine. The findings in this study could provide useful insight toward the quality control of passion fruit wine.

## 1. Introduction

Passion fruit (*Passiflora edulis*) is a sweet and seedy fruit that originates from Brazil [1]. It has been reported to contain various nutrients and micronutrients, such as vitamins and polyphenols. These bioactive ingredients have been confirmed to lower the incidence of chronic diseases [1]. Passion fruit can be freshly consumed or processed into juice in the food market. However, fresh passion fruit is readily spoiled during the storage period [2]. Fermenting passion fruit into passion fruit wine could preserve its nutritional value, enhance its unique aromatic feature and perpetuate its shelf life period, which could further improve the economic value of passion fruit.

The volatiles composition in fruit wine is mainly derived from three sources. The wine maceration process results in the release of fruit volatiles into wine, which provides wine with a fruit varietal aromatic feature [3]. Volatile compounds formed during the wine aging process could enhance the wine aroma complexity through introducing aged flavor and aroma in wine [4]. The wine fermentation process accumulates fermentative note volatile compounds in wine, and yeast play a vital role in altering the fermentative volatile compounds in fruit wine [3].

It has been reported that yeast mainly metabolizes sugar to alcohol and carbon dioxide through a series of enzymatic metabolisms during the fermentation process [5]. Meanwhile, yeast-derived aromatic compounds could be released into wine along with the metabolism of yeast on nutrients [3]. Different yeast strains possess different metabolic enzymes, which could incorporate different fermentative volatiles in wine during fermentation and further result in an alteration in the overall aroma of fruit wine [6]. For example, fermentation with different Saccharomyces cerevisiae strains result in a different volatile compounds composition in the wine of grape, pineapple, mulberry and goji berry [6,7,8]. The volatile composition of passion fruit has been investigated using gas chromatography–mass spectrometry (GC-MS) and gas chromatography–olfactory (GC-O) [2]. However, no studies have been conducted to investigate the volatile composition of passion fruit wine and its aromatic features to the best of our knowledge. Sulfur volatile compounds have been confirmed to play important roles in the tropical aromas of passion fruit [9]. Sulfur volatiles in tropical fruit have been widely analyzed using gas chromatography coupled with sulfur-selective detectors, such as chemiluminescence sulfur detectors and pulsed flame photometric detectors [10]. The chemiluminescence sulfur detector lacks a high detection sensitivity on sulfur compounds, whereas sulfur compounds analyzed by a pulsed flame photometric detector can only be measured under a limited concentration range [11,12]. Gas chromatography–Orbitrap–mass spectrometry (GC-Orbitrap-MS) has gained much attention in identifying and quantifying trace volatiles in the food science area since Orbitrap-MS possesses a better mass accuracy and relative ion abundance consistency under a wide concentration range of sulfur volatiles [13,14]. Therefore, GC-Orbitrap-MS and GC-quadrupole mass spectrometry (GC-qMS) were used in the present study for the analysis of sulfur volatiles and other volatile compounds in passion fruit wine, respectively.

Due to the complexity of fruit wine aroma, chemical analysis alone cannot comprehensively evaluate the impact of varied yeast strains on the aroma of passion fruit wine. Electronic nose (E-nose) is a common method used to differentiate the aromatic variation according to volatile compounds, whereas sensory evaluation is normally used for aromatic feature measurement using professional panelists [15,16,17]. Both E-nose and sensory evaluation were further conducted in this study to unveil the aromatic features of passion fruit wines fermented under different yeast strains. Partial least squares regression (PLSR) analysis was performed to further establish the correlation between volatile compounds and sensory attributes in these passion fruit wines under different yeast strains, which could elucidate the effect of yeast strains on the alteration in the aroma in passion fruit wine. 

At present, there are very few studies on the aroma composition of passion fruit wine, and it is unclear whether and to what extent different yeast will affect the aroma quality and volatile composition of passion fruit wine. In this study, firstly, we analyzed the physical and chemical indexes of the four passion fruit wines. Secondly, we adopted E-nose to preliminarily analyze the aroma differences in different wine. Thirdly, qualitative and quantitative analyses of the constituents were employed by GC-qMS and GC-Orbitrap-MS. Finally, the contribution of volatile compounds to the different aroma profiles was explored by sensory evaluation and PLSR analysis. Our findings can provide useful insight toward the improvement and control of passion fruit wine quality.

## 2. Materials and Methods

### 2.1. Chemicals and Reagents

Four commercial yeast strains, ES488, CY3079, BV818 and VIC, were chosen because of their steady fermentation characteristics and excellent flavor enhancement according to the manufacturer instructions. The strains ES488 and CY3079 *(Saccharomyces cerevisiae*) were purchased from Enartis (Tracete, Italy) and Lallemand (Montréal, QC, Canada), respectively. Strains BV818 and VIC (*Saccharomyces bayanus*) were provided from Angel Yeast Co., Ltd. (Hubei, China). Pectinase and potassium metabisulfite were purchased from Enartis (Tracete, Italy). Water was purified through a Milli-Q purification system (Millipore, Bedford, MA, USA). Folin-Ciocalteu reagent was purchased from Beijing Sulaibao Tech Co., Ltd. (Beijing, China). Sodium hydroxide, sodium chloride, ammonium sulfate, aluminum nitrate, sodium sulfate, sodium carbonate, tartaric acid and ethanol were of analytical grade and purchased from Beijing Chemical Works (Beijing, China). The volatile standards used in this study were purchased from Sigma-Aldrich (St. Louis, MO, USA). These standards had a purity above 95%. The internal standard (4-methyl-2-pentanol) was obtained from Sigma-Aldrich (St. Louis, MO, USA) with a 98% purity.

### 2.2. Passion Fruit Wine Fermentation

Commercially ripened passion fruit (*Passiflora edulis*) samples were harvested in 2016 from Guangxi, China. The fruits (approximately 40 kg) were treated as follows: washing, removal of shell, deseeding and pressing. The obtained juice was approximately 10 L, then mixed with 10 L distilled water in a 20 L capacity stainless steel tank, followed by being treated with 45 mg/L sulfur dioxide and 3 g/hL pectinase (Enartis, Italy). The resultant juice was adjusted to pH of 3.4 and sugar content of 180 g/L using tartaric acid and sucrose, respectively. Erlenmeyer flasks (2000 mL) containing 1500 mL of the resultant juice were equipped with fermentation locks containing aqueous sulfuric acid solution. The fermentation of the passion fruit juice was initiated by adding 20 g/hL of a yeast strain (ES488, CY3079, BV818 or VIC) and maintained at 17 ± 1 °C under static conditions. The reducing sugar content of the seedless passion fruit juice was monitored to reflect the fermentation performance (Appendix A). When the relative density was not altered in a three-consecutive-day period, the fermentation was terminated by adding 40 mg/L potassium metabisulfite at 0 °C. Each fermentation was carried out in duplicate. 

### 2.3. Physicochemical Indices and Flavonoid Content

The analysis of physicochemical indices of the passion fruit wine, including the residual sugar content, pH, free sulfur content, alcohol content, volatile acidity and titratable acidity, followed the Chinese Standard: Analytical methods of wine and fruit wine (GB 15038-2006). The total flavonoid content in the passion fruit wine was analyzed according to published methods with minor modifications [18]. In brief, the passion fruit wine sample (0.5 mL) was mixed with 0.4 mL of 5% *w/v* NaNO_2_ solution. The resultant mixture was kept at room temperature for 6 min and then mixed with 0.4 mL of 10% *w/v* Al(NO_3_)_3_ solution. Subsequently, the mixture was mixed with 0.4 mL of 1.0 mol/L NaOH solution and then brought up to 25 mL using distilled water. The resultant mixture was incubated at room temperature for 15 min and then measured at 510 nm using the UV/Vis spectrophotometer (UNICO, Shanghai, China). The total flavonoid content of the wine sample was expressed as mg rutin equivalents per liter of wine (mg rutin/L). Each measurement was carried out in triplicate. 

### 2.4. Electronic-Nose (E-Nose) Analysis

E-nose has been widely applied for organoleptic properties of fruit wines due to its easy measurement ability, lower cost and high reliability [19]. The mechanism behind the E-nose analysis is that volatile molecules transported through clean environment air to the metal oxide semiconductor sensor could result in oxide reduction reactions on the sensor surface. This could further alter the electric resistance of the sensors to collect signal [19]. The E-nose analysis of these passion fruit wine samples was carried out using a WMA portable Electronic Nose 3 (Airsense Analytics Inc., Schwerin, Germany) consisting of a multiple-gas-sensor array, a signal-collecting unit and pattern recognition software [16]. The sensors array contained 10 metal oxide semiconductor-type chemical sensors, including MOS1 (aromatic), MOS2 (broadrange), MOS3 (aromatic), MOS4 (hydrogen), MOS5 (arom-aliph), MOS6 (broad-methane), MOS7 (sulfur-organic), MOS8 (broad-alcohol), MOS9 (sulph-chlor) and MOS10 (methane-aliph) [16]. The wine sample (1.0 mL) was placed in a 10.0 mL sealed glass vials and equilibrated at 40 °C for 30 min under agitation. The clean environmental air was used as carrier gas to transport volatiles in the headspace of the sealed glass vials to the sensor chamber with controlled humidity and temperature at a flow rate of 300 mL/min during measurement [19]. The change in conductivity in sensor array is expressed by the ratio of conductivity G/G0, where G and G0 represent the conductance of the sensor after and before exposure to the gas sample, respectively. Each measurement cycle lasted for 100 s, which makes the sensor reach a stable state, and the interval of data acquisition was 1 s using a computer. Between the measurement cycle, the sensor was cleaned for 5 min with cleaning gas filtered through activated charcoal to return the sensor signal to the baseline. The software package Win Muster (v.1.6.2) can be bundled with electronic nose instruments to computerize measurement and data collection.

### 2.5. Aroma Compounds Analysis 

#### 2.5.1. Extraction of Volatile Compounds

Headspace solid phase micro-extraction (HS-SPME) was used to extract volatile compounds of the passion fruit wine samples according to the published methods with minor modifications [20,21]. The wine sample (5.0 mL) was mixed with 10 µL of 1.0018 mg/L 4-methyl-2-pentanol (internal standard) and 1.00 g NaCl in a 15 mL vial containing a magnetic stirrer. The vial was tightly capped with a PTFE-silicon septum and equilibrated at 40 °C for 30 min under agitation. An SPME fiber (50/30 µm DVB/Carboxen/PDMS, Supelco, Bellefonte, PA, USA) was inserted into the headspace of the vial to adsorb volatiles at 40 °C under the same agitation for 30 min. Afterwards, the fiber was removed from the vial headspace and immediately inserted into the GC injector for the desorption of volatile compounds at 250 °C for 25 min. Each sample was carried out in duplicate. 

#### 2.5.2. Gas Chromatography-Orbitrap-Mass Spectrometry (GC-Orbitrap-MS)

A Thermo Scientific Trace 1300 gas chromatography coupled with a Thermo Scientific Q-Exactive Orbitrap mass spectrometer (GC-Orbitrap MS, Thermo Scientific, Bremen, Germany) was used to analyze sulfur volatile compounds in these passion fruit wine samples. A DB-WAX capillary column (30 m × 0.25 mm, 0.25 µm thickness, J&W Scientific, Folsom, CA, USA) was used to separate sulfur volatiles under a carrier gas (helium) flow rate at 1 mL/min. The oven temperature gradient was programed as follows: 40 °C held for 2 min, then ramped from 40 °C to 230 °C at 4 °C/min and then 230 °C held for 15 min. A 70 eV voltage ionization energy was used in Orbitrap-MS under an electron impact mode with a mass scanning range of 30 to 3000 amu under a selective ion mode. The retention indices were carried out using a C7-C24 n-alkane series (Supelco, Bellefonte, PA, USA) under the same chromatographic conditions. Sulfur volatiles with available standards were identified through comparing their mass spectrum and retention indices with NIST11 database and their reference standard. Sulfur volatiles without available reference standard were tentatively identified through comparison of their mass spectrum with NIST11 database.

For quantitation, a synthetic passion fruit wine model containing 9 g/L tartaric acid, 12% *v/v* ethanol and 10 g/L glucose was prepared. The model solution followed the same extraction and GC-Orbitrap MS analysis as the passion fruit wine samples. Sulfur volatiles were quantified using their corresponding reference standard when the reference standards were available. For sulfur volatiles without available standards, they were quantified using the standard, with their chemical structure or carbon atoms similar to the volatiles [21,22]. Each sample was analyzed on GC-Orbitrap MS in duplicate.

#### 2.5.3. Gas Chromatography-Quadrupole Mass Spectrometry (GC-qMS)

An Agilent 6890 gas chromatograph coupled with an Agilent 5975 mass spectrometry (Agilent Technologies, Santa Clara, CA, USA) was used to separate volatile compounds in the passion fruit wine according to a published method [21]. An HP-INNOWAX capillary column (60 m × 0.25 mm, 0.25 µm thickness, J&W Scientific, Folsom, CA, USA) was used to separate volatile compounds through a split-less inlet mode under a flow rate of helium (carrier gas) at 1 mL/min. The gradient of oven temperature was programed as follows: 50 °C held for 1 min, then increased to 220 °C at a 3 °C/min rate, held at 220 °C for 5 min, then increased to 250 °C at a 5 °C/min rate and finally held 250 °C for 5 min. The electron impact mode was used in mass spectrometer with a voltage of 70 eV ionization energy. The mass scan was carried out in a range of *m/z* 20 to 450 under a selective ion mode. Retention indices were calculated using a C7-C24 n-alkane series (Supelco, Bellefonte, PA, USA) under the same chromatographic condition. Volatile compounds with commercially available standards were identified by comparing their mass spectrum and retention indices with the NIST11 library and their reference standard. Volatile compounds without the reference standards were tentatively identified through comparing their mass spectrum and retention index with the library of NIST11 [23]. 

For quantitation, the volatile standards were dissolved into a passion fruit wine model solution consisting of 12% *v/v* ethanol, 10 g/L glucose and 9 g/L tartaric acid. The model solution was extracted using the same extraction procedure as the passion fruit wine samples and analyzed under the same chromatographic conditions. The volatile compounds with the reference standard were quantified using the reference standard, whereas the quantitation of the volatiles without the available standard was carried out by the standard that possessed the similar carbon atoms numbers or chemical structure [21,22,24]. Each sample was analyzed on GC-qMS in duplicate.

#### 2.5.4. Odor Activity Value

Odor activity value (OAV) is a term used to indicate the contribution of a volatile compound to the overall aroma of fruit wine [21]. The OAV of a volatile compound was calculated by its concentration in wine over its hydro-alcoholic odor threshold. A volatile compound with its OAV above 1 indicated that this volatile could significantly contribute its flavor notes to the overall aroma of fruit wine. Additionally, the overall aroma was described using a seven-aroma series that included floral, fruity, fatty, chemical, caramel, earthy and herbaceous aromas. Each aroma was calculated through summing up the OAVs of the volatiles with this aromatic feature.

### 2.6. Sensory Evaluation

Sensory attributes of these passion fruit wine samples were evaluated using a professional panel consisting of 14 panelists (6 males and 8 females) with an age range between 21 and 35 years. These panelists had basic knowledge on food sensory evaluation and had been trained using a 54-aroma kit (Le Nez du Vin^®^, Carnoux-en-Provence, France) for at least one month until these panelists exhibited a less than 5% deviation in assessing each aroma trait in the aroma kit [25]. Sensory evaluation took place in sensory laboratory that complied with international standards, and was performed in duplicate. A total of 30 milliliters of passion fruit wines was presented in a 215 mL Bordeaux glass and distributed in a completely randomized order. Each sample was evaluated for 10 min. Between samples, the panelists were asked to have 5 min rest in order to minimize fatigue. 

Every panelist was asked to describe the passion fruit wine aroma using the terminology of aroma kit, and to rank their intensity using an 11-point scale (none to very intensive, from 0 points to 10 points). Aroma characteristics were quantized with the geometric mean (GM%), which referred to a mixture of intensity and frequency of detection. It was calculated using square root of the product of detection frequency (F%) and average intensity (I%) [17,26]. According to ISO 11035, wine descriptors with low GM% could be eliminated as the featured descriptors in wine, whereas descriptors with a GM% higher than 20% were used for subsequent analysis. Five sensory attributes, namely passion fruit, mango, green apple, lemon and floral aromas, provided the vocabulary for further descriptive analysis.

### 2.7. Statistical Analysis

Principal component analysis (PCA) algorithm was used to distinguish passion fruit wine fermented by different yeasts based on electronic nose signals, and loading analysis was used to determine the contribution of the sensor. Sensors stabilized after 85 s and before 95 s. Thus, the data from 85 s and 95 s were chosen as the input of PCA analysis. Data points were selected for analysis every 2 s, and 5 data points were used for PCA analysis in a single test. Each wine sample was carried out on the E-nose analysis three times. 

Data were expressed as the mean ± standard deviation of duplicate tests. One-way analysis of variance (ANOVA) was used to compare the difference among the means using Tukey’s multiple range test under SPSS Statistical 21.0 software (SPSS Inc., Chicago, IL, USA). A difference was considered as significant at a 0.05 level. A partial least-squares regression (PLSR) model was carried out using SIMCA-P (Umetrics). Principal component analysis (PCA), based on autoscaling the original data, was performed using the MetaboAnalyst (http://www.metaboanalyst.ca/, accessed on 4 September 2018).

## 3. Results and Discussion

### 3.1. Physicochemical Indices and Total Flavonoids

These passion fruit wines fermented with the different strains exhibited a similar alcohol and total sulfur content (Table 1). The VI and BV-fermented passion wines contained the highest reducing sugar content, followed by the wine fermented by the ES strain. The CY-strain-fermented passion fruit wine exhibited the lowest reducing sugar content. Additionally, these passion fruit wines showed a similar level regarding the total titratable acidity, volatile acidity and pH value. Flavonoids are the major antioxidants in fruit wine, and their level plays a vital role in the nutritional and sensory quality of fruit wine [8]. In the present study, the CY-fermented wine possessed the highest total flavonoids content, whereas the lowest total flavonoids level was found in the BV wine.

### 3.2. E-Nose 

In the present study, the aromatic feature of these wine samples was analyzed under 10 chemical sensors under the E-nose analysis, and their aromatic differences were analyzed using principal component analysis (Figure 1) [23,24]. The first and second principal component represented 95.8% and 2.6% of the total variance, respectively. An obvious segregation among these passion fruit wine samples was observed in the scores plot (Figure 1A). For instance, the CY and ES-strain-fermented passion fruit wines were positioned at the positive side of the PC1. However, the position of these two strain-fermented wines at the PC2 separated these two wine samples from each other (the CY wine at the negative side of the PC2 and the ES wine at the positive side of the PC2). Similarly, the BV-strain-fermented wine was situated at the positive side of the PC2 and near the zero point of the PC1. The VI-fermented passion fruit wine resided at the left corner of the score plot (the negative side of both PC1 and PC2). In the loading plot (Figure 1B), the MOS8 (broad-alcohol) and MOS6 (broad-methane) signals exhibited a primary contribution on the PC1. Meanwhile, the signals derived from the MOS2 (broadrange), MOS7 (sulfur-organic), MOS9 (sulph-chlor) and MOS1 (aromatic) also contributed their effect to the PC1. 

### 3.3. Sulfur Compounds Analysis Using GC-Orbitrap-MS

Sulfur compounds have been considered as critical volatile compounds that primarily determine the featured aroma of tropic fruits, although these compounds exist in fruits with a trace amount level [9]. In the present study, a total of 17 sulfur compounds were detected in passion fruit wine samples (Table 2, Appendix A).

Different yeast strains resulted in a variation in the sulfur compounds composition and concentration in the passion fruit wine samples. Among these compounds, methionol was the most prominent sulfur compounds in all samples and was reported as the most abundant sulfur compound in wine [27], and can confer odors of cauliflower and potato at a concentration above its perception threshold (1000 μg/L) [9]. Although methionol was found in a variety of tropical fruit juice [10], the formation of methionol in wine is clearly related to the metabolism of L-methionine during alcohol fermentation [28]. As shown in Table 2, the content of methionol was significantly higher in the sample fermented by ES488 than other yeasts; this suggested that the methionine metabolism routes of the strain ES488 were more robust than others in passion fruit wine. It has been reported that 3-mercaptohexanol (3MH) is a featured aromatic compound in passion fruit [10]. Its precursor, cysteine conjugate, exists in fruit juice and can be cleaved into 3MH through enzymes in yeast during alcoholic fermentation in wine [29]. 3MH has been reported to further interact with acetic acid through the esterification reaction in fruit wine to yield 3-mercaptohexyl acetate (3MHA), another important sulfur volatile responsible for passion fruit’s featured aroma [29]. 3MH significantly contributed its flavor notes to the overall aroma in these passion fruit wines due to its high odor activity value (OAV) in the current study (Table 2). Meanwhile, the passion fruit wine fermented by ES488 exhibited a lower level of 3MH than the other wine samples. Regarding 3MHA, only ES488-fermented wine was found to contain this sulfur compound, but there is very limited information on the clear correlations between 3MH and 3MHA in the passion fruit matrix, and additional research is required. 2-methyl-4-propyl-1,3-oxathiane (MPO), another sulfur compound, provides the most characteristic passion fruit odor and occurs naturally in passion fruit juice, especially in the yellow fruits [10]. It was recently identified and quantitated in wine, and has a close relation to 3MH during wine fermentation and ageing [30]. As shown in Table 2, MPO was found in all samples; meanwhile, the passion fruit wine fermented by CY3079 had a significantly lower level, which was more than seven times lower than the others. On the basis of OAV, it seemed that the contribution of MPO to the aroma of CY3079 wine was not significant, but, with its highest content of 3MH, it seems necessary to conduct further studies to better elucidate the sensory impact on the passion fruit wine aroma according to the research of [30]. It has been reported that 3-(methylthio) propyl acetate exhibits herbaceous and vegetable-like odor notes [9], and that these passion fruit wine samples possess different levels of 3-(methylthio) propyl acetate (Table 2). Similarly, 3-(Methylthio)propanoic acid ethyl ester can significantly contribute its fruity and cheese-like scents to the overall aroma of these passion fruit wines. Its concentration in these passion fruit wine samples showed a significant difference. Benzothiazole and 2-methyltetrahydrothiophen-3-one were also found in these samples, with their OAV above 1, indicating that their chemical scents (rubber and natural gas) could be significantly incorporated into the overall aroma of the passion fruit wine. It is worth noting that the passion fruit wine fermented by VIC contained a more abundant such sulfur compound composition than the other samples, whereas strains BV818 and CY3079 enhanced the accumulation of 3MH in passion fruit wine compared to VIC and ES488.

### 3.4. Volatile Compounds Analysis Using GC-qMS

A total of 78 volatile compounds were detected in these passion fruit wine samples, including 32 esters, 17 terpenes, 8 alcohols, 7 acids, 6 norisoprenoids, 2 ketones, 2 benzenes, 2 phenols, 1 furan compound and 1 lactone volatile compound (Table 3). Among these volatile compounds, esters, terpenes and alcohols were found to be the dominant volatiles in these wine samples. These strain-fermented wine samples possessed a similar volatiles composition. However, the concentration of the volatile compounds showed differences in these wine samples.

#### 3.4.1. Esters

Esters are mainly produced through the esterification between alcohols and acids during wine fermentation, and these volatiles have been confirmed to play vital roles in the determination of the wine’s overall aroma [3]. Esters can be divided into ethyl esters, acetate esters and other esters according to their chemical nature [39]. Fatty acid metabolism can result in the formation of acyl-CoA, and acyl-CoA can be further converted into ethyl esters through ethanolysis [40]. These passion fruit wines were rich in ethyl esters (Table 3). Ethyl acetate has been reported to be a dominant ethyl ester in fruit wine, and its concentration directly affectes the overall aroma of fruit wine. For example, the aroma complexity of wine could be enhanced with an ethyl acetate level below 150 mg/L in wine, whereas a concentration higher than 150 mg/L could bring an unpleasant aroma to wine [41]. In the present study, these passion fruit wine samples exhibited a similar ethyl acetate concentration, and its levels in these wine samples were all above its threshold, but below 150 mg/L. This indicated that ethyl acetate could provide these wine samples with fruity and floral scents. The other dominant individual ethyl esters in these wine samples included ethyl acetate, ethyl butanoate, ethyl hexanoate, ethyl caprylate, ethyl caprate, ethyl 9-decenoate and ethyl cinnamate. These ethyl esters had their OAV above 1, indicating that they significantly contributed to the overall aroma of these passion fruit wines. These volatiles were reported to bring fruity and floral aromatic notes [25,31]. For example, ethyl caprylate possessed sweet, floral, fruity and banana scents, whereas ethyl butanoate and ethyl hexanoate had apple, cherry and banana notes [41]. Among these wine samples, the VI and CY-fermented wine exhibited the highest concentration of these dominant ethyl esters. The BV wine appeared to show the lowest content of these ethyl esters, indicating that the BV-fermented passion fruit wine might lack a floral and fruity aroma compared to the other strain-fermented wine samples. It has been reported that ethyl butanoate was the main component of the passion fruit aroma and showed the highest odoriferous importance for the characteristic aroma of passion fruit [2]. In our study, there was a significant difference in the content of ethyl butanoate among these passion fruit wine samples. The CY wine exhibited the highest concentration on ethyl butanoate, followed by the ES and VI wine. The BV-fermented passion fruit wine showed the lowest content of ethyl butanoate. It should be pointed out that some ethyl esters (such as ethyl laurate in the present study) with an OAV between 0.1 and 1 could also contribute their fruity flavor character to fruit wine due to the synergistic effect in the wine matrix [20,25]. The rest of the ethyl esters found in these passion fruit wine samples had their concentration below their threshold, which indicated that they might not contribute to the overall aroma of the wine, although a concentration difference existed in these strain-fermented wines [42].

Regarding acetate esters, these wine samples fermented by different strains contained different acetate ester contents (Table 3). Isopentyl acetate appeared to be the major ethyl acetates in these wine samples. Isopentyl acetate was reported to exhibit banana flavor notes [21,35]. In the present study, the concentration of isopentyl acetate was much higher than its threshold in these wine samples, indicating that its flavor scents significantly contributed to the overall aroma of the passion fruit wines. It should be noted that the concentration difference was also found in some other acetate esters among these wine samples. However, their level in wine was not above their threshold, limiting their contribution to the overall aroma in the passion fruit wine.

It was observed that these wine samples contained 10 other esters (Table 3), and some of these esters showed a concentration difference among the wine samples. However, their low concentration (OAV below 0.1) resulted in a limited contribution to the wine’s overall aroma.

#### 3.4.2. Alcohols

Amino acids could be metabolized under the enzymatic activity of yeast strains during the wine fermentation process to yield higher alcohols (fusel alcohols) [3]. Higher alcohols have been confirmed to affect the overall aroma of wine. It has been reported that the presence of higher alcohols in wine with a low concentration (lower than 300 mg/L) could favor the overall aroma of wine through enhancing the aromatic complexity, whereas higher alcohols with a high concentration (higher than 400 mg/L) could result in an off-flavor of the wine aroma [6]. In this study, three wine samples showed a total higher alcohol concentration above 400 mg/L, which might result from the massive consumption of amino acids by these yeast strains during fermentation [21]. Isobutanol and isopentanol were found to be the dominant higher alcohols in these wine samples, and their concentration represented approximately 90% of the total fusel alcohol content (Table 3). Isopentanol is normally produced through deamination and decarboxylation from leucine, whereas the transamination of valine could result in the formation of isoleucine during the wine fermentation process. Isopentanol and isobutanol both possess solvent and alcohol scents [21,34,40]. These wine samples showed a significant difference in the isobutanol level. The highest and lowest concentrations of isobutanol were found in the VI and CY wine, respectively. In addition, 2-heptanol was found to have a similar concentration in these wine samples. Isopentanol exhibited the lowest content in the BV-fermented passion fruit wine sample (Table 3). This fusel alcohol was also detected in grape wines, with little flavor contribution due to its low concentration [21,38]. However, its moldy, mushroom and fruity aroma could significantly contribute to the passion fruit wine aroma since its concentration was higher than its threshold in these wine samples. Phenyl ethyl alcohol had the highest level in the CY wine, followed by the VI and BV wine samples. The ES-fermented wine sample exhibited the lowest phenyl ethyl alcohol concentration. Phenyl ethyl alcohol possesses sweet and rose scents and its floral notes could be fully incorporated into the overall aroma of the passion fruit wines due to its high OAV. Benzyl alcohol is described as a roasted and sweet aroma [21,35]. The CY and BV passion fruit wine exhibited the highest and lowest level of benzyl alcohol. Its aromatic contribution in the passion fruit wine was limited since its concentration was below the threshold in these wine samples.

#### 3.4.3. Acids

Fatty acids can be degraded to produce volatile acids during wine fermentation, and different yeast strains have been reported to regulate the metabolism of fatty acids under wine fermentation. As a result, the volatile acid composition could be altered in different strain-fermented wines [39]. In the present study, seven volatile acids were detected in these passion fruit wines (Table 3). These wine samples contained a similar content of the total volatile acids. It has been reported that volatile acids might induce an unpleasant aroma in wine when their level is higher than their threshold [3]. In the present study, six volatile acids were found to have a higher concentration than their threshold in these passion fruit wine samples. This indicated that the unpleasant fatty, cheesy, pungent and rancid flavor might be incorporated into the overall aroma of these passion fruit wines [21,35]. It should be noted that these passion fruit wine samples exhibited a significant difference in the level of octanoic acid, isovaleric acid, hexanoic acid and 9-decenoic acid. These fatty acids could enhance the rancid, cheesy and fatty aromas in the passion fruit wine due to their high OAV. For example, the CY passion fruit wine showed the highest concentration of octanoic acid, whereas its lowest concentration was found in the ES-fermented wine. The other volatile acids did not show a concentration difference in these different strain-fermented passion fruit wine samples.

#### 3.4.4. Ketones

Ketones, a group of carbonyl compounds, are mainly synthesized through unsaturated fatty acids metabolism in wine fermentation. Carbonyl compounds are considered as the key components for improving the elegance, uniqueness and richness of the wine aroma [39]. In the present study, the CY-fermented passion fruit wine possessed the highest content of the total ketones. These passion fruit wine samples exhibited a low level of ketones (Table 3). Such a low level of ketones in these wines might have resulted from their conversion to alcohols during wine fermentation [3]. All of these ketones present in these passion fruit wine samples had an OAV below 1.

#### 3.4.5. Benzenes and Phenols

The major benzenes found in these strain-fermented passion fruit wine samples included styrene and benzaldehyde (Table 3). These benzenes’ concentration in these wines was below their odor threshold, indicating that their flavor features might not significantly be incorporated into the overall aroma in passion fruit wine. Phenols are yielded through the metabolisms of phenolic acids during wine fermentation, and these compounds can be further metabolized during the wine aging period [42]. It has been reported that the phenols composition is mainly determined by the fruit genotype and these compounds could play a role in improving the wine aroma under low concentration [3]. In this study, 4-vinylguaiacol and 2,4-di-tert-butylphenol were the only two phenols found in these wine samples. These phenols could introduce perfume aromatic scents to these passion fruit wines.

#### 3.4.6. Terpenes and Norisoprenoids

Terpenes have been confirmed to be one of the most important aromatic compounds in fruit wine since these volatiles possess a relatively low odor threshold. Terpenes could provide wine with fruity, rose, floral and sweet aromas [21]. In this study, these wine samples were found to contain 17 terpenes (Table 3). Among these terpenes, β-myrcene, α-terpineol, linalool, citronellol, geranyl acetone, cis-rose oxide, trans-rose oxide and geraniol appeared to be the dominant individual terpenes in the passion fruit wines. β-Myrcene was reported to possess a lychee flavor and its concentration was much higher than its odor threshold in these passion fruit wine samples, indicating that its flavor notes could significantly contribute to the passion fruit wine aroma [9]. These different strain-fermented wine samples contained different β-myrcene concentrations. The higher linalool concentration was found in the CY and VI-fermented wines, whereas the BV and ES wines contained a lower linalool level. The lowest linalool concentration was found in the BV-fermented passion fruit wine. It should be noted that the concentrations of linalool in these wine samples were all higher than their threshold. This suggested that its floral and musky aromas were significantly incorporated to produce the finest expression of the passion fruit wine. Additionally, these wine samples exhibited different α-terpineol concentrations, and their concentrations were much higher than their odor threshold. The VI-fermented passion fruit wine contained the highest level of α-terpineol, followed by the ES and CY wine. The BV passion fruit wine showed the lowest content. α-Terpineol was described as having lilac, floral and sweet notes [25]. It should be noted that nerolidol, 4-terpineol and nerol showed concentration differences in these different strain-fermented wines. However, their low concentration restricted their flavor notes being incorporated into the overall aroma of these passion fruit wines [36].

Similar to terpenes, norisoprenoids also possess low odor thresholds in fruit wine, and these volatiles can always enhance the wine aroma with flower and fruit notes. Based on their structure, norisoprenoids can be divided into megastigmane and non-megastigmane norisoprenoids [3]. In the present study, β-ionone and α-ionone were two major megastigmane norisoprenoids in these wine samples (Table 3). The concentration of β-ionone was significantly higher than its odor threshold in these passion fruit wines. These different strain-fermented passion fruit wines contained a comparable β-ionone concentration. Therefore, rose and violet could be substantially incorporated into the passion fruit wine aroma [21,37]. Similarly, the flavor feature of α-ionone could also be enhanced in these wines since its OAV was above 1 in these wine samples. α-Ionone and β-ionone exhibited the highest level in the CY passion fruit wine.

#### 3.4.7. Other Volatiles

One furan (2,3-dihydrobenzofuran) and one lactone (butyrolactone) were found in these passion fruit wine samples (Table 3). The CY passion fruit wine contained the highest level of 2,3-dihydrobenzofuran, whereas the lowest level was found in the BV fermented wine. These passion fruit wines contained a similar level of lactone. The aromatic contribution of lactone was limited in these passion fruit wines due to its low concentration [38].

### 3.5. Aroma Profile

The aromatic profile of these passion fruit wine samples was evaluated using the aroma series features. Each aroma feature was calculated by summing up the OAV of each volatile compound with the same aromatic note and its OAV above 1 (Table 3). These passion fruit wines fermented by different yeast strains were featured with strong fruity, floral, herbaceous, caramel, chemical, earthy and fatty aromas, with a lack of roasted, nutty and spice scents (Appendix A).

The floral aroma played a dominant role in the aroma profile of these passion fruit wines. The major floral aroma contributors included one alcohol (phenyl ethyl alcohol), four esters (ethyl cinnamate, ethyl caprylate, ethyl acetate and phenethyl acetate), seven terpenes (α-terpineol, linalool, citronellol, geranyl acetone, cis-rose oxide, trans-rose oxide and geraniol), and three norisoprenoids (β-damascone, β-ionone and TDN). It is worth noting that other fruit wines, such as mulberry and Airen macerated wines, exhibited a fruity aroma as the dominant aromatic feature as well [8,20,43]. Among these passion fruit wine samples, the CY wine exhibited the highest floral aroma intensity, followed by the VI and ES wine. The least floral aroma was found in the BV wine. Besides the floral aroma, the passion fruit wines were also abundant in fruity and fatty aromas. The fruity aroma in the passion fruit wines was primarily determined by 20 volatile compounds, including 2-heptanol, ethyl butanoate, ethyl hexanoate, ethyl caprylate, ethyl caprate, ethyl 9-decenoate, ethyl acetate, ethyl laurate, isopentyl acetate, isobutyl acetate, phenethyl acetate, hexyl butyrate, isoamyl caprylate, β-myrcene, β-damascone, α-ionone, ethyl 3-(methylthio)propionate, 3-mercaptohexanol, 3-mercaptohexyl acetate and 2-methyl-4-propyl-1,3-oxathiane. The varietal thiols, such as 3-mercaptohexanol and 3-mercaptohexyl acetate, with desirable citrus and tropical fruit flavor notes, were of particular importance to the fruity aroma of the passion fruit wines [9]. The ES-fermented passion fruit wine exhibited the highest intensity in the fruity aroma since 3-mercaptohexyl acetate was only found in the ES wine with an extremely high OAV (1332.66).

The fatty aroma in these passion fruit wines was mainly attributed to isovaleric acid, hexanoic acid, octanoic acid, decanoic acid, 9-decenoic acid, lauric acid, isopentanol, ethyl laurate and isoamyl caprylate. The ES-fermented passion fruit wine sample was found to possess the lowest intensity in the fatty aroma. The caramel aroma in these passion fruit wine samples mainly consisted of one alcohol (isopentanol), one ester (isoamyl caprylate), two terpenes (α-terpineol and linalool) and one norisoprenoid (β-damascone). These strain-fermented passion fruit wines displayed a similar intensity in the caramel aroma. The chemical aroma in these passion fruit wines mainly resulted from the accumulation of fusel alcohols (higher alcohols). It has been reported that isobutanol and isopentanol primarily determined the chemical flavor in Cabernet Sauvignon wines [21]. In the present study, these two fusel alcohols were the critical chemical aroma contributors in these passion fruit wines. These different strain-fermented passion fruit wines also showed a different intensity in the chemical aroma, which mainly resulted from the content difference in 2-methyltetrahydrothiophen-3-one in these wines.

The herbaceous flavor has been reported to be mainly determined by the alcohols’ presence in wine, and this aromatic note was normally considered as a negative factor in the overall amora in wine [23]. In the present study, isobutanol was the main alcohol to bring a herbaceous aroma to the passion fruit wines. This volatile compound had a higher concentration than its odor threshold in the ES and VI passion fruit wines, leading the ES and VI wines to have strongest herbaceous aroma. However, the CY and BV-fermented passion fruit wines did not display such an aromatic feature due to the low OAV of isobutanol. 3-(Methylthio)propyl acetate was reported to provide a herbaceous aroma [9]. The CY passion fruit wine sample displayed the lowest intensity in the herbaceous aroma. According to the aromatic profile, the CY and ES passion fruit wines were found to exhibit strong floral and fruity aromas, whereas the ES and BV-fermented wines were distinguished by their strong herbaceous flavor.

### 3.6. Sensory Evaluation

A professional panel was conducted to evaluate the sensory attributes of these passion fruit wine samples, and the aromatic profiles of each passion fruit wine was described using the terminologies in the aroma kit. The aromatic features of these passion fruit wine samples were assessed using the geometric mean (GM%). It has been accepted that the aromatic feature with a GM% higher than 20% could be considered as a distinguished aroma [25]. In the present study, five aromatic characteristics with a GM% above 20% were selected for further analysis, including passion fruit, mango, green apple, lemon and floral aromas (Figure 2). These different strain-fermented passion fruits possessed different aromatic characteristics (Appendix A). For instance, the CY-fermented passion fruit wine exhibited the strongest note on the mango, green apple and floral aromas, whereas the BV passion fruit wine had passion fruit and lemon scents as the featured flavors. It has been reported that 3-mercaptohexanol and 3-mercaptohexyl acetate played an essential role in the contribution of the passion fruit aroma [44]. In the current study, the CY and ES passion fruit wine possessed the highest level of 3-mercaptohexanol and 3-mercaptohexyl acetate, respectively, among these wine samples. However, the CY passion fruit wine was rated as having the least passion fruit note by the panelists. The mango aromatic feature in these passion fruit wines was mainly attributed to ethyl caprylate, ethyl caprate, β-myrcene and ocimene [36]. The esters mainly strengthened the fruity character in the passion fruit wines, whereas the floral aroma in these wines was mainly from alcohol volatiles, including linalool, nerol, limonene, α-terpenol and citronellol. Phenyl ethyl alcohol and phenethyl acetate have a rose-like flavor. They formed in the alcoholic fermentation in yeasts and were produced from L-phenylalanine through the Ehrlich pathway. In our study, we found that phenyl ethyl alcohol and phenethyl acetate mainly contributed to a floral aromatic feature in passion fruit wine. It is worth noting that the overall aroma perception of these passion fruit wine samples judged by the panelists resulted from the flavor contribution of each major volatile and the contribution from the synergistic effects of multiple volatile compounds in these wines.

### 3.7. Correlation between Volatiles and Sensory Attributes in Passion Fruit Wines Using PLSR

Partial least squares regression analysis (PLSR) was further utilized in the present study to elucidate the correlation between the volatile compounds (OAV > 1 for volatiles or OAV > 0.1 for esters) and sensory features (GM% > 20%) in these passion fruit wine samples (Figure 3). The correlation loading in the PLSR analysis included 44 volatiles determined by GC-Orbitrap-MS and GC-qMS (Table 4) and 5 aromatic features (passion fruit, mango, green apple, lemon and floral). It was found that the floral note was positively correlated with phenyl ethyl alcohol and phenethyl acetate (Figure 3). Both phenyl ethyl alcohol and phenethyl acetate were described as the rose flavor note. The biosynthesis of phenyl ethyl alcohol and phenethyl acetate is performed mainly by yeasts (Saccharomyces cerevisiae) through the L-phenylalanine biotransformation system, which is mainly concentrated in the deep fermentation process. Phenyl ethyl alcohol and phenethyl acetate are obtained in two forms, where each has its own advantages and disadvantages. One was obtained by chemical synthesis; however, these processes tend to produce undesirable by-products and give off odors. The other was extracted from the essential oil contained in flowers such as rose or jasmine, but their low concentration makes the recycling process complicated and expensive. In this study, phenyl ethyl alcohol and phenethyl acetate were produced by alcohol fermentation. Similarly, a positive correlation of the mango descriptor in these passion fruit wines was established with octanoic acid, ethyl 9-decenoate, TDN and benzothiazole. Terpenoids and esters, such as cis-rose oxide, β-damascone, isopentyl acetate and ethyl butanoate, exhibited a positive correlation with the green apple aroma in these wine samples. The passion fruit descriptor in these wines was strongly correlated with sulfur volatiles, especially 2-methyltetrahydrothiophen-3-one, 2-methyl-4-propyl-1,3-oxathiane and 3-(methylthio)propyl acetate, whereas 9-decenoic acid appeared to exhibit a correlation with the lemon sensory attribute in these passion fruit wines.

## 4. Conclusions

In conclusion, the passion fruit wines fermented with four different commercial yeast strains exhibited different aromatic features using E-nose analysis. GC-Orbitrap-MS and GC-qMS detected 17 and 78 volatile compounds in these passion fruit wines, with 44 volatile compounds as the vital volatiles for the aromatic contribution to the overall aroma in passion fruit wines due to their high odor activity value. The sensory evaluation indicated that these passion fruit wines exhibited passion fruit, mango, green apple, lemon and floral aromas as their major flavor characteristics. The CY3079 − fermented passion fruit wine had the strongest note in the mango, green apple and floral aromas, whereas the BV818 passion fruit wine exhibited passion fruit and lemon scents. Partial least squares regression analysis suggested that the featured aromas of fruity, mango and green apple in these passion fruit wines were mainly attributed to sulfur volatiles, esters and terpenes, and terpenes, respectively.

## Figures and Tables

**Figure 1 foods-11-03789-f001:**
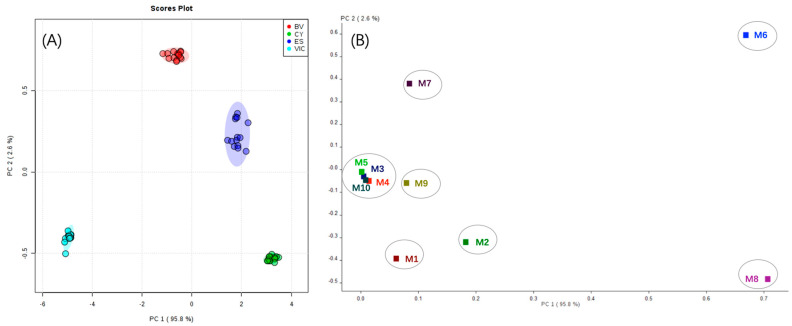
Principal component analysis of passion fruit wine fermented by commercial yeast strain ES488 (ES), BV818 (BV), VIC (VI) and CY3079 (CY). (**A**) The scores plot of passion fruit wine samples and (**B**) the loadings analysis of different metal oxide semiconductor (MOS) − type chemical sensors: 1 (aromatic), 2 (broad range), 3 (aromatic), 4 (hydrogen), 5 (arom − aliph), 6 (broad − methane), 7 (sulfur − organic), 8 (broad − alcohol), 9 (sulph − chlor) and 10 (methane − aliph).

**Figure 2 foods-11-03789-f002:**
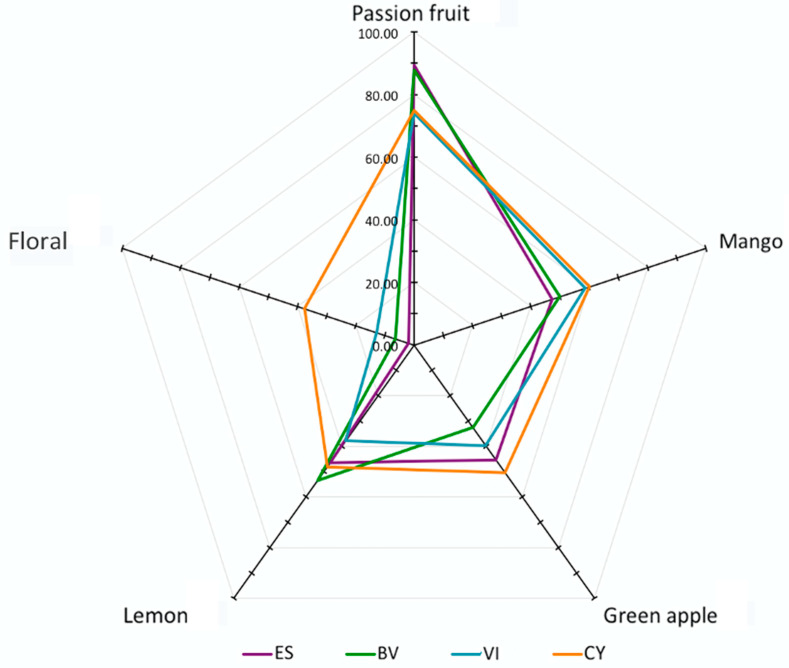
Sensory attributes of passion fruit wine fermented by commercial yeast strain ES488 (ES), BV818 (BV), VIC (VI) and CY3079 (CY) using a professional panel. The geometric mean (GM%) was used to assess the aromatic features of these passion fruit wine samples.

**Figure 3 foods-11-03789-f003:**
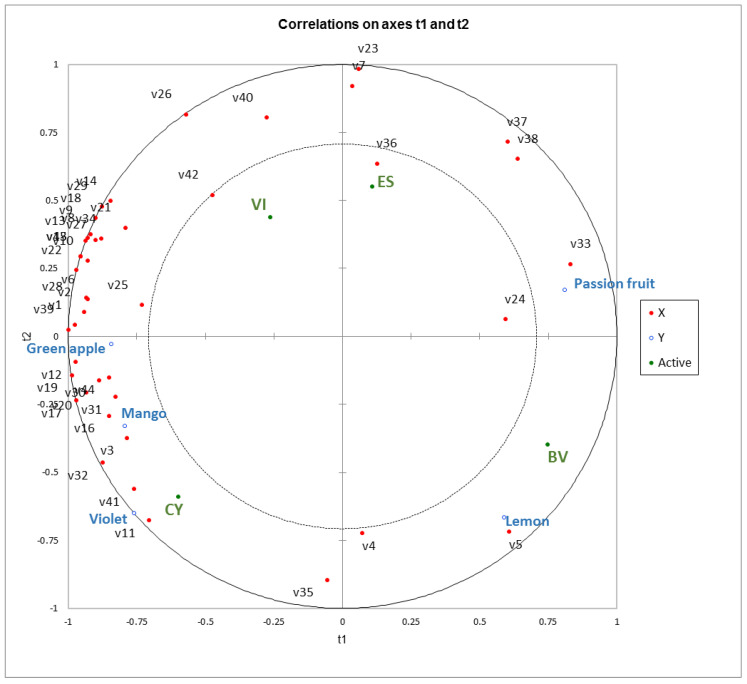
Partial least squares regression (PLSR) analysis using volatile compounds (OAV > 1 or esters OAV > 0.1, as listed in Table 4) and aroma descriptors (passion fruit, mango, green apple, lemon and floral with GM% > 20%) in passion fruit wine fermented by commercial yeast strain ES488 (ES), BV818 (BV), VIC (VI) and CY3079 (CY).

**Table 1 foods-11-03789-t001:** Physicochemical indices and total flavonoids content in passion fruit wine fermented by commercial yeast strain ES488 (ES), BV818 (BV), VIC (VI) and CY3079 (CY).

Parameters	Yeast Strain			
	ES	BV	VI	CY
Residual sugar (g/L)	3.43 ± 0.47 ab	5.2 ± 0.98 a	4.31 ± 0.49 a	1.57 ± 0.04 b
Total titratable acidity (g/L)	9.84 ± 0.50 a	8.54 ± 0.42 a	9.69 ± 0.05 a	9.48 ± 0.33 a
Volatile acidity (g/L)	0.10 ± 0.01 a	0.08 ± 0.01 a	0.13 ± 0.09 a	0.08 ± 0.01 a
Total sulfur (mg/L)	199.6 ± 79.76 a	82.4 ± 10.18 a	204.0 ± 112.01 a	211.6 ± 3.96 a
Alcohol content (% *v/v*)	9.35 ± 0.74 a	9.63 ± 0.45 a	9.15 ± 0.66 a	9.31 ± 0.39 a
pH	3.15 ± 0.04 a	3.34 ± 0.11 a	3.19 ± 0.03 a	3.23 ± 0.04 a
Total flavonoids (mg Rutin/L)	179.81 ± 9.69 ab	148.29 ± 4.26 b	189.96 ± 6.2 ab	213.26 ± 24.03 a

Data are the mean ± standard deviation of duplicate tests. Different letters in each row indicate significant difference at a significant level of 0.05.

**Table 2 foods-11-03789-t002:** Sulfur volatile compounds in passion fruit wine fermented by commercial yeast strain ES488 (ES), BV818 (BV), VIC (VI) and CY3079 (CY).

Sulfur Volatile	RI ^a^	Id ^b^	Standard	HRF Score ^c^	Qi ^d^	Odor Threshold (μg/L)	Odor Descriptor	Odor Series ^e^	Concentration (µg/L)			
									ES	BV	VI	CY
** *Heterocycle* **												
**Benzothiazole**	1945	A	Benzothiazole	99.6281	135.0137	0.05 [29]	Rubber [29]	7	n.d.	n.d.	0.41 ± 0.03 b	0.88 ± 0.03 a
4-Methyl-5-vinylthiazole	1507	B	Benzothiazole	98.0454	125.0294	NF ^f^	Fatty, roasted, nutty [9]	8,9	2.64 ± 0.44 a	n.d.	3.02 ± 0.02 a	n.d.
Thiazole	1246	B	Benzothiazole	98.4352	84.9981	38 [28]	Popcorn, peanut [28]	8	n.d.	1.21 ± 0.03 b	1.56 ± 0.30 b	4.47 ± 0.52 a
**MTHT ^g^**	1547	A	MTHT	100	116.0291	0.09 [29]	Natural gas, sulfurous [29]	7	8.55 ± 0.48 c	14.25 ± 0.11 a	11.22 ± 0.31 b	2.96 ± 0.19 d
**2-Methyl-4-propyl-1,3-oxathiane**	1478	C	Diethyl disulfide	99.3228	145.0683	7.1 [30]	Passion fruits [9]	1	22.41 ± 0.97 b	22.48 ± 0.98 b	25.64 ± 0.14 a	2.76 ± 0.25 c
2-Methyl-1,3-oxathiane	1130	C	Diethyl disulfide	97.0532	103.0213	NF			n.d.	n.d.	1.10 ± 0.00 a	n.d.
** *Ester sulfide* **												
**3MPAEE ^h^**	1553	A	3MPAEE	94.9905	74.0185	1 [31]	Fruity, cheese like [9]	1,8	10.34 ± 0.27 b	n.d.	9.88 ± 0.11 b	11.46 ± 0.35 a
Ethyl 3-(methylthio)-(E)-2-propenoate	1721	B	3MPAEE	95.4488	99.9978	NF	Sweet, sour, faintly fruity [9]	1	0.31 ± 0.06 a	0.37 ± 0.00 a	0.26 ± 0.02 a	0.34 ± 0.01 a
**3-(Methylthio)propyl acetate**	1623	A	3-(Methylthio)propyl acetate	95.1033	88.0341	0.115 [31]	Herbaceous, cabbage [9]	3	88.01 ± 2.71 a	53.06 ± 0.52 b	32.74 ± 0.47 c	n.d.
** *Alkyl sulfide* **												
Methionol	1707	A	Methionol	97.0737	106.0447	1000 [29]	Cauliflower, potato [29,32]	3	696.57 ± 42.93 a	317.22 ± 15.29 b	190.53 ± 0.61 c	216.59 ± 11.55 c
** *Polyfunctional thiol* **												
**3-Mercaptohexanol**	1848	A	3-Mercaptohexanol	93.3314	82.0777	0.06 [28]	Passion fruit, mango [29]	1	5.51 ± 0.22 b	9.79 ± 0.89 a	8.12 ± 0.72 ab	10.06 ± 1.23 a
3-Mercaptohexyl hexanoate	2015	B	3-Mercaptohexyl acetate	94.5746	87.0263	NF	Passion fruit [9]	1	n.d.	n.d.	127.82 ± 10.88 a	87.00 ± 3.84 b
**3-Mercaptohexyl acetate**	1723	A	3-Mercaptohexyl acetate	99.7294	87.0263	0.004 [29]	Passion fruit [28]	1	5.33 ± 0.59 a	n.d.	n.d.	n.d.
** *Alkyl disulfide* **												
Diisopropyl disulfide	1254	B	Diethyl disulfide	93.0231	108.0062	NF	Oniony, durian-like [9]	3	n.d.	0.42 ± 0.01 b	0.60 ± 0.05 b	3.46 ± 0.23 a
** *Thioacid* **												
Ethanethiolic acid	1168	C	S-Ethyl ethanethioate	94.0675	43.0178	NF	Cooked, roasted, meaty [33]	9	n.d.	115.86 ± 4.78 a	104.89 ± 3.25 b	n.d.
** *Others* **												
[2-(Ethylsulfanyl)ethyl]benzene	2133	C	Diethyl disulfide	92.8947	75.0262	NF			n.d.	n.d.	4.13 ± 0.69 b	13.10 ± 0.57 a
3-(Methylthio)-2-butanone	1832	C	Diethyl disulfide	94.6535	75.0262	NF	Milky, creamy, potato [33]		n.d.	n.d.	n.d.	0.41 ± 0.01 a

^a^ Retention indices on DB-wax column, ^b^ Identification of the compounds: ‘A’ means identified by mass spectrum and RI agree with standards, ‘B’ means tentatively identified by mass spectrum, agrees with the mass spectral database and RI agrees with literature, ‘C’ means high-resolution filtering score and mass spectral database. ^c^ HRF (high-resolution filtering score); percentage of the spectrum obtained by MS Orbitrap that can be explained by combination of accurate mass, library matching and percentage of explained ions observed. ^d^ Quantitative ion. ^e^ 1 = fruity, 2 = floral, 3 = herbaceous (or vegetal), 4 = nutty, 5 = caramel, 6 = earthy, 7 = chemical, 8 = fatty, 9 = roasted. ^f^ NF: not found. ^g^ 2-Methyltetrahydrothiophen-3-one. ^h^ 3-(Methylthio)propanoic acid ethyl ester. Data are the mean ± standard deviation of duplicate tests. ‘n.d.’: not detected. Different letters indicate significant differences among the results at a significant level of 0.05. Bold indicates the significant contribution of the compound to aroma (OAV > 1). Odor descriptors and reported odor threshold mentioned in the listed literatures: [9] (Hui et al., 2010); [28] (Landaud et al., 2008); [29] (Waterhouse et al., 2016); [30] (Chen et al., 2018); [31] (v.Gemert et al., 2011–2019); [33] Descriptors were retrieved from http://www.thegoodscentscompany.com (accessed on 13 August 2018).

**Table 3 foods-11-03789-t003:** Concentration, odor threshold, odor descriptor and odor series of individual volatile compounds in passion fruit wine fermented by commercial yeast strain ES488 (ES), BV818 (BV), VIC (VI) and CY3079 (CY).

Volatile Compound	RI ^a^	Id ^b^	QI ^c^	Odor Threshold (μg/L)	Odor Descriptor	Odor Series ^d^	Concentration (µg/L)			
							ES	BV	VI	CY
** *Acids* **							205,040.38 ± 29,134.24 b	245,714.78 ± 814.30 a	256,799.48 ± 24,490.93 a	264,654.2 ± 31,529.96 a
Isobutyric acid	1583	B	43	2300 [8]	Rancid, butter, cheese [21]	8 [21]	1855.58 ± 139.18 ab	1213.38 ± 33.54 c	2193.52 ± 767.73 a	1660.49 ± 453.03 b
**Isovaleric acid**	1681	B	60	3000 [8]	Acid, rancid [21]	8 [8]	4575.16 ± 97.86 b	3029.86 ± 147.04 c	4673.86 ± 1805.20 b	5564.63 ± 415.15 a
**Hexanoic acid**	1873	B	60	420 [8]	Rancid, cheese [8]	8 [21]	13,975.82 ± 1650.66 ab	11,381.84 ± 114.94 b	18,900.83 ± 5529.80 a	18,450.31 ± 4669.58 a
**Octanoic acid**	2063	A	60	500 [21]	Rancid, cheese, fatty [21]	8 [8]	54,407.14 ± 5030.83 c	58,769.71 ± 93.62 bc	70,996.10 ± 4769.74 ab	74,275.61 ± 9003.27 a
**Decanoic acid**	2262	A	73	1000 [21]	Rancid, fatty [21,25]	8 [21]	118,779.57 ± 20,029.80 a	157,164.30 ± 744.56 a	147,790.98 ± 9579.09 a	150,567.30 ± 18,696.09 a
**9-Decenoic acid**	2311	B	55	1000 [8]	Fatty, rancid [34]	7, 8 [8]	3810.33 ± 603.71 c	8689.53 ± 327.34 a	4923.08 ± 4175.12 bc	5759.31 ± 403.45 b
**Lauric acid**	2489	B	73	1000 [35]	Fatty, laurel oil [25,33]	8	7636.78 ± 668.34 a	5466.16 ± 247.9 a	7321.11 ± 873.65 a	8376.55 ± 1746.57 a
** *Alcohols* **							472,952.96 ± 9518.25 a	330,484.89 ± 1267.20 b	516,453.63 ± 7080.15 a	457,145.87 ± 24,568.51 a
**Isobutanol**	1091	A	43	75,000 [34]	Alcohol, solvent, green, [21]	3, 7 [21]	88,912.60 ± 559.51 b	62,793.03 ± 571.42 c	109,425.09 ± 38,823.43 a	44,033.30 ± 9690.94 d
**Isopentanol**	1217	A	55	60,000 [34]	Solvent, sweet, alcohol, nail polish [21]	7, 5, 8 [21]	379,943.26 ± 9252.87 a	263,678.39 ± 1550.53 b	401,399.34 ± 145,317.11 a	403,736.49 ± 30,555.55 a
**2-Heptanol**	1327	A	45	200 [21]	Fruity, fresh lemon, moldy [21,33]	1, 6 [21]	430.90 ± 29.02 a	366.99 ± 2.01 a	438.26 ± 106.82 a	440.98 ± 7.23 a
**2-Nonanol**	1526	B	45	58 [34]	Green [25]	7	211.20 ± 3.20 a	166.48 ± 2.19 b	210.25 ± 69.06 a	220.94 ± 0.22 a
1-Octanol	1566	A	56	800 [21]	Jasmine, lemon [21]	2 [21]	120.37 ± 4.16 a	87.46 ± 0.81 b	111.13 ± 31.66 a	117.91 ± 0.42 a
1-Decanol	1773	B	55	400 [21]	Cucumber [25]	1	37.15 ± 0.43 a	24.55 ± 0.04 b	38.71 ± 13.36 a	45.01 ± 6.62 a
**Phenyl ethyl Alcohol**	1927	B	91	1100 [36]	Sweet, rose [34]	2 [34]	2522.42 ± 2209.19 d	2741.48 ± 53.32 c	3989.52 ± 1765.75 b	7542.04 ± 27.79 a
Benzyl alcohol	1893	A	79	200,000 [34]	Roasted, sweet [34]	5, 9 [8]	775.06 ± 40.91 b	626.51 ± 18.54 c	841.33 ± 306.15 b	1009.20 ± 5.95 a
** *Ethyl esters* **							50,281.38 ± 1724.87 ab	34,347.23 ± 20.92 b	55,152.07 ± 10,237.29 a	53,018.60 ± 3717.68 a
**Ethyl acetate**	702	A	43	12,000 [34]	Pineapple, fruity [25]	1, 2 [8]	40,827.05 ± 2532.88 a	28,313.83 ± 71.91 b	43,994.58 ± 15,457.33 a	42,468.69 ± 5270.44 a
**Ethyl butanoate**	941	A	71	400 [8]	Fruity, pineapple [21]	1 [21]	2191.65 ± 225.06 b	1732.94 ± 23.20 c	2238.81 ± 849.70 b	2639.77 ± 3.76 a
**Ethyl hexanoate**	1236	A	88	80 [21]	Banana, green apple [21]	1 [34]	1618.28 ± 44.39 a	1010.71 ± 16.03 b	1862.32 ± 793.27 a	1807.05 ± 343.99 a
Ethyl 3-hexenoate	1318	A	41	Nf ^e^	Sweet, fruity [9]		38.59 ± 0.72 a	26.57 ± 0.47 b	39.49 ± 14.90 a	40.53 ± 2.95 a
Ethyl heptanoate	1342	A	88	220 [25]	Fruity [25]	1	1.78 ± 0.03 a	0.99 ± 0.12 b	1.85 ± 0.01 a	n.d.
Ethyl 2-hexenoate	1361	B	55	NF			45.25 ± 0.69 ab	36.48 ± 0.28 b	36.57 ± 14.81 b	54.77 ± 1.67 a
**Ethyl caprylate**	1456	A	88	580 [8]	Sweet, floral, fruity, banana [21]	1, 2 [21]	4667.01 ± 345.55 b	2475.57 ± 28.00 c	6050.95 ± 147.55 a	4982.68 ± 285.94 b
Ethyl 3-hydroxybutyrate	1529	B	43	2000 [21]	Grape [21]	1 [21]	39.74 ± 4.38 b	31.28 ± 1.01 c	45.71 ± 16.42 b	54.73 ± 0.87 a
**Ethyl caprate**	1656	A	88	200 [21]	Fruit, fatty, mango notes [8,9]	1 [8]	192.96 ± 1.42 a	169.11 ± 0.76 b	198.20 ± 20.53 a	200.55 ± 14.59 a
Ethyl 3-hydroxyhexanoate	1691	B	71	NF	Floral, passion fruit, fruity [9]		30.67 ± 1.30 c	24.19 ± 0.25 d	36.83 ± 13.36 b	40.51 ± 2.75 a
**Ethyl 9-decenoate**	1705	B	55	100 [35]	Fruity [25]	1	263.84 ± 3.43 b	233.92 ± 0.41 d	252.28 ± 0.49 c	299.16 ± 3.1 a
Ethyl laurate	1860	A	88	1500 [21]	Oily, fatty, fruity [21,25]	1, 8 [21]	336.58 ± 0.68 c	269.63 ± 1.35 d	365.95 ± 0.78 b	399.61 ± 0.69 a
Ethyl tetradecanoate	2067	B	88	2000 [35]	Mild waxy, soapy [25]	7	4.29 ± 0.01 ab	3.60 ± 0.18 b	4.47 ± 1.36 ab	5.27 ± 0.31 a
Ethyl palmitate	2263	B	88	1500 [35]	Fatty, rancid, fruity, sweet [25]	1, 8	9.11 ± 1.37 a	5.45 ± 0.29 b	8.03 ± 2.17 ab	8.53 ± 1.64 ab
Ethyl 9-hexadecenoate	2280	B	55	NF			5.48 ± 0.66 a	4.45 ± 0.20 a	6.20 ± 1.12 a	6.05 ± 1.50 a
**Ethyl cinnamate**	2141	B	131	1.1 [37]	Flowery, balsamic [37]	2 [37]	9.10 ± 0.06 c	8.51 ± 0.06 d	9.83 ± 1.21 b	10.70 ± 0.52 a
** *acetate esters* **							10,342.79 ± 71.35 bc	8103.97 ± 50.87 c	12,836.75 ± 2030.97 ab	15,100.01 ± 644.26 a
Isobutyl acetate	867	A	43	1600 [6]	Waxy, fruity, apple, banana [6]	1, 7	389.22 ± 1.26 b	291.36 ± 151.36 b	532.67 ± 165.58 a	293.97 ± 126.53 b
**Isopentyl acetate**	1138	A	43	160 [34]	Banana [21]	1 [21]	9246.77 ± 64.21 bc	7264.25 ± 3276.98 c	11,625.75 ± 3686.05 ab	13,500.07 ± 2032.27 a
Hexyl acetate	1283	A	43	1500 [21]	Apple, cherry, pear, floral [34]	1, 2 [21]	114.70 ± 5.40 b	93.30 ± 43.47 b	140.98 ± 51.28 a	141.71 ± 14.41 a
Octyl acetate	1486	B	43	NF			119.13 ± 1.82 a	109.02 ± 14.01 b	121.49 ± 11.89 a	121.08 ± 2.93 a
Benzyl acetate	1745	B	108	NF			1.79 ± 0.01 b	1.29 ± 0.70 c	1.93 ± 0.70 b	2.33 ± 0.22 a
Phenethyl acetate	1834	A	104	1800 [8]	Fruity, rose [8]	1, 2 [34]	471.18 ± 1.31 b	344.75 ± 197.22 b	413.93 ± 214.33 b	1040.85 ± 119.62 a
** *other esters* **							697.63 ± 6.22 b	586.07 ± 1.30 c	700.70 ± 13.76 b	800.10 ± 25.02 a
Isobutyl hexanoate	1361	B	99	NF	Sweet, fruity [33]		95.15 ± 0.19 a	94.50 ± 0.51 a	104.66 ± 66.82 a	108.60 ± 1.71 a
Methyl octanoate	1401	A	74	200 [8]	Intense citrus [34]	1 [21]	12.63 ± 0.54 a	6.95 ± 0.09 b	14.69 ± 5.92 a	14.30 ± 4.03 a
Hexyl butyrate	1424	B	43	250	Sweety, fruity	1	93.43 ± 0.01 a	92.87 ± 0.04 a	96.46 ± 1.11 a	93.60 ± 0.35 a
Isopentyl hexanoate	1471	A	70	1000 [34]	Pineapple, cheese [8]	1 [34]	23.71 ± 0.77 a	11.27 ± 0.11 b	24.67 ± 12.11 a	24.08 ± 5.29 a
2-Heptyl hexanoate	1593	B	43	NF			10.93 ± 0.21 a	6.93 ± 0.17 b	8.62 ± 3.25 ab	9.71 ± 0.83 ab
Methyl decanoate	1609	B	74	1200 [34]	Waxy, soap, fruity [25]	1, 7	7.00 ± 0.47 a	4.61 ± 0.11 a	7.27 ± 1.73 a	6.74 ± 1.79 a
isoamyl caprylate	1671	A	70	125 [8]	Sweet, fruity, cheese, cream [34]	1, 5, 8 [34]	21.22 ± 0.47 a	13.42 ± 0.24 b	22.94 ± 6.71 a	23.70 ± 4.77 a
Diethyl succinate	1687	A	101	1,200,000 [34]	Fruity, melon [8]	1 [34]	164.85 ± 5.53 b	122.87 ± 1.40 c	156.59 ± 51.74 b	236.20 ± 4.34 a
Isoamyl decanoate	1882	A	70	NF	Waxy [25]		83.01 ± 1.11 a	75.35 ± 0.22 a	85.94 ± 9.27 a	88.96 ± 8.08 a
Methyl salicylate	1798	B	120	40 [25]	Mint [25]	3	21.82 ± 0.03 a	16.43 ± 0.07 a	19.51 ± 7.69 a	25.43 ± 1.67 a
** *Phenols* **							23.09 ± 0.30 a	19.25 ± 0.09 b	23.32 ± 0.16 a	24.62 ± 1.65 a
4-Vinylguaiacol	2209	B	150	1100 [6]	Smoky, bacon, phenolic [38,33]	9	5.26 ± 0.03 c	4.37 ± 0.04 d	5.63 ± 0 b	6.31 ± 0.03 a
2,4-Di-tert-butylphenol	2304	B	191	200 [35]	Phenolic [33]		17.83 ± 0.02 b	14.88 ± 0.05 c	17.69 ± 0.01 b	18.31 ± 0.07 a
** *Benzene* **							1526.42 ± 88.45 a	1462.96 ± 0.75 a	1479.58 ± 39.05 a	1288.11 ± 9.77 b
Styrene	1274	A	104	NF	Sweet [33]		234.19 ± 18.71 a	160.22 ± 26.25 b	229.02 ± 102.73 a	229.21 ± 2.64 a
Benzaldehyde	1543	B	77	2000 [8]	Roasted, almond [8]	4, 9 [34]	1292.23 ± 107.32 a	1302.74 ± 0.74 a	1250.56 ± 162.80 ab	1058.90 ± 24.83 b
** *Ketone* **							n.d.	4.71 ± 0.29 b	6.64 ± 0.08 b	67.91 ± 0.04 a
Sulcatone	1350	B	43	NF	Citrus, green [33]		n.d.	3.92 ± 0.08 b	n.d.	60.49 ± 0.01 a
Nonanal	1409	A	57	15 [8]	Green, slightly pungent [34]	3 [34]	n.d.	0.79 ± 0.00 c	6.64 ± 0.08 b	7.42 ± 0.05 a
** *Terpenes* **							8072.81 ± 19.46 c	6368.65 ± 10.67 d	9381.48 ± 30.26 a	8886.08 ± 18.24 b
**β-Myrcene**	1178	A	43	15 [36]	Fruity, herbal [9]	1, 3	220.82 ± 2.04 c	207.49 ± 0.58 d	286.52 ± 3.44 b	308.58 ± 0.87 a
d-Limonene	1209	A	68	200 [8]	Flowery, green, citrus [34]	2, 3 [34]	6.09 ± 0.21 c	10.81 ± 0.14 b	5.98 ± 0.03 c	16.25 ± 0.42 a
β-Ocimene	1271	B	93	NF	Floral, herb [33]		103.62 ± 0.48 c	97.13 ± 0.26 d	165.63 ± 3.17 b	191.41 ± 0.71 a
Terpinolen	1298	A	93	NF	Citrus, lemon [25]		67.36 ± 0.79 c	65.90 ± 0.75 c	178.61 ± 4.20 a	143.11 ± 0.30 b
Neo-allo-ocimene	1389	B	121	NF			38.23 ± 0.28 c	36.47 ± 0.13 c	59.77 ± 0.03 b	92.55 ± 1.62 a
4-Terpineol	1616	A	71	40-110 [35]	Earth, musty [33]	6	35.76 ± 0.11 c	28.66 ± 0.69 d	42.95 ± 0.35 b	45.03 ± 0.43 a
**α-Terpineol**	1710	A	59	250 [34]	Lilac, floral, sweet [8]	2, 5 [34]	1862.68 ± 0.79 c	1520.89 ± 9.80 d	2469.29 ± 11.10 a	2078.17 ± 4.18 b
**Linalool**	1555	A	71	25 [8]	Floral, musk [21]	2, 5 [21]	4658.75 ± 10.61 c	3574.84 ± 3.84 d	5121.81 ± 15.30 b	5199.80 ± 15.55 a
**Citronellol**	1779	A	69	100 [21]	Rose [8]	2 [8]	529.80 ± 5.96 b	348.28 ± 0.83 c	553.72 ± 5.94 a	267.34 ± 3.31 d
Nerol	1814	A	69	500 [34]	Violets, floral [34]	2 [21]	178.44 ± 1.76 c	137.47 ± 0.12 d	201.89 ± 4.13 b	219.53 ± 1.58 a
Nerolidol	1483	A	55	700 [38]	Rose, apple, green, citrus [38]	1, 2	96.67 ± 0.30 a	94.97 ± 0.11 b	96.26 ± 0.19 a	96.97 ± 0.25 a
**Geranyl acetone**	1871	B	43	60 [8]	Floral [8]	2 [34]	201.33 ± 2.46 a	181.77 ± 1.65 b	126.90 ± 0.73 d	153.31 ± 2.69 c
**cis-Rose oxide**	1355	A	139	0.2 [34]	Lychee [34]	2 [21]	3.96 ± 0.01 a	3.41 ± 0.01 c	3.64 ± 0.02 b	4.00 ± 0.05 a
**trans-Rose oxide**	1372	A	139	0.2 [21]	Lychee [21]	2 [21]	2.34 ± 0.01 a	2.21 ± 0.01 c	2.37 ± 0.01 a	2.28 ± 0.00 b
**Geraniol**	1859	A	69	20 [21]	Citric, geranium, passion fruit [8,9]	2 [21]	64.91 ± 0.68 ab	56.46 ± 0.30 c	63.90 ± 0.25 b	65.45 ± 0.21 a
*cis*-furan linalool oxide	1452	A	59	500 [25]	Rose, wood [25]	2	1.05 ± 0.01 c	0.99 ± 0.00 d	1.16 ± 0.01 b	1.19 ± 0.00 a
*trans*-furan linalool oxide	1483	A	59	500 [35]	Rose, wood [25]	2	1.00 ± 0.01 b	0.90 ± 0.01 c	1.08 ± 0.01 a	1.11 ± 0.00 a
** *Norisoprenoids* **							1305.18 ± 29.62 a	1012.91 ± 3.41 c	1243.59 ± 5.89 b	1384.59 ± 46.73 a
**β-Damascone**	1835	A	69	0.14 [8]	Sweet, exotic flowers, stewed [21]	1, 2, 5 [21]	3.91 ± 0.00 b	3.70 ± 0.05 b	4.26 ± 0.12 a	4.20 ± 0.05 a
**α-Ionone**	1859	B	121	2.6 [21]	Sweet fruit [21]	1 [21]	23.80 ± 0.37 a	18.16 ± 0.11 b	24.86 ± 1.63 a	24.08 ± 0.19 a
**β-Ionone**	1956	A	177	0.09 [21]	Rose, violet [34]	2 [37]	785.67 ± 28.06 b	640.02 ± 2.49 c	739.73 ± 2.93 b	901.39 ± 48.85 a
**TDN ^g^**	1765	B	157	20 [25]	Pleasant, flowery, petrol [25]	2,7	36.38 ± 0.82 b	30.59 ± 0.09 c	33.5 ± 1.5 bc	41.09 ± 1.94 a
Edulan I	1627	B	177	NF			389.16 ± 0.65 a	271.43 ± 0.25 d	375.15 ± 1.84 b	351.25 ± 1.90 c
Edulan II	1501	B	177	NF			66.26 ± 0.20 a	49.01 ± 0.22 c	66.09 ± 0.61 a	62.58 ± 0.16 b
** *Furan* **							262.05 ± 17.98 b	194.59 ± 12.77 c	277.42 ± 124.76 b	350.99 ± 5.30 a
2,3-Dihydrobenzofuran	2356	B	120	NF			262.05 ± 17.98 b	194.59 ± 12.77 c	277.42 ± 124.76 b	350.99 ± 5.30 a
** *other components* **							38.40 ± 13.66 a	30.10 ± 5.60 a	46.60 ± 20.08 a	23.58 ± 3.31 a
Butyrolactone	1645	B	42	2000 [21]	Sweet, fruity, toasty, caramel [21]	1, 5, 9 [21]	38.40 ± 13.66 a	30.10 ± 5.60 a	46.60 ± 20.08 a	23.58 ± 3.31 a

^a^ Retention indices on HP-Innowax column. ^b^ Identification of the compounds: ‘A’ means identified by mass spectrum and RI agree with standards, ‘B’ means tentatively identified by mass spectrum, agrees with the mass spectral database and RI agrees with literature. ^c^ Quantitative ion. ^d^ 1 = fruity, 2 = floral, 3 = herbaceous (or vegetal), 4 = nutty, 5 = caramel, 6 = earthy, 7 = chemical, 8 = fatty, 9 = roasted. e NF: not found. ^f^ Ethyl 2-hydroxy-4-methylpentanoate. ^g^ Dehydro-ar-ionene. Data are the mean ± standard deviation of duplicate tests. ‘n.d.’: not detected. Different letters indicate significant differences among the results at a significant level of 0.05. Bold indicates the significant contribution of the compound to aroma (OAV > 1). Odor descriptors and reported odor threshold mentioned in the listed literatures [8] (Ouyang et al., 2017); [9] (Hui et al., 2010); [21] (Cai et al., 2014); [25] (Wang et al., 2017); [35] (Tao et al.,2010); [34] (Liu et al., 2018); [38] (Peng et al., 2013); [36] (Pino et al., 2011); [37] (Yuan et al., 2016); [33]. Descriptors were retrieved from http://www.thegoodscentscompany.com (accessed on 13 August 2018).

**Table 4 foods-11-03789-t004:** Volatile compounds in passion fruit wine fermented by commercial yeast strain ES488 (ES), BV818 (BV), VIC (VI) and CY3079 (CY) selected for partial least squares regression analysis (volatiles with OAV > 1 and esters with OAV > 0.1).

Compound Number	Volatile Compound	Odor Series	ES	BV	VI	CY
1	**Isovaleric acid**	8	1.53	1.01	1.56	1.85
2	**Hexanoic acid**	8	33.28	27.1	45	43.93
3	**Octanoic acid**	8	108.81	117.54	141.99	148.55
4	**Decanoic acid**	8	118.78	157.16	147.79	150.57
5	**9-Decenoic acid**	7, 8	3.81	8.69	4.92	5.76
6	**Lauric acid**	8	7.64	5.47	7.32	8.38
7	**Isobutanol**	3, 7	1.19	0.84	1.46	0.59
8	**Isopentanol**	7, 5, 8	6.33	4.39	6.69	6.73
9	**2-Heptanol**	1, 6	2.15	1.83	2.19	2.2
10	**2-Nonanol**	7	3.64	2.87	3.62	3.81
11	**Phenyl ethyl alcohol**	2	0.87	2.49	3.63	6.86
12	**Ethyl butanoate**	1	5.48	4.33	5.6	6.6
13	**Ethyl hexanoate**	1	20.23	12.63	23.28	22.59
14	**Ethyl caprylate**	1,2	8.05	4.27	10.43	8.59
15	**Ethyl caprate**	1	0.96	0.85	0.99	1
16	**Ethyl 9-decenoate**	1	2.64	2.34	2.52	2.99
17	**Ethyl cinnamate**	2	8.27	7.74	8.94	9.73
18	**Ethyl acetate**	1, 2	3.4	2.36	3.67	3.54
19	**Isopentyl acetate**	1	57.79	45.4	72.66	84.38
20	**β-Myrcene**	1, 3	14.72	13.83	19.1	20.57
21	**α-Terpineol**	2, 5	7.45	6.08	9.88	8.31
22	**Linalool**	2, 5	186.35	142.99	204.87	207.99
23	**Citronellol**	2	5.3	3.48	5.54	2.67
24	**Geranyl acetone**	2	3.36	3.03	2.11	2.56
25	***cis*-Rose oxide**	2	19.8	17.03	18.2	20.01
26	***trans*-Rose oxide**	2	11.7	11.07	11.85	11.41
27	**Geraniol**	2	3.25	2.82	3.2	3.27
28	**β-Damascone**	1, 2, 5	27.94	26.46	30.44	30.02
29	**α-Ionone**	1	9.15	6.98	9.56	9.26
30	**β-Ionone**	2	8729.67	7111.36	8219.21	10,015.43
31	**TDN**	2, 7	1.82	1.53	1.67	2.05
32	**Benzothiazole**	7	0	0	8.28	17.6
33	**2-Methyltetrahydrothiophen-3-one**	7	95.02	158.31	124.65	32.88
34	**Ethyl 3-(methylthio)propionate**	1	10.34	0	9.88	11.46
35	**3-Mercaptohexanol**	1	91.83	163.19	135.26	167.71
36	**3-Mercaptohexyl acetate**	1	1332.66	0	0	0
37	**2-Methyl-4-propyl-1,3-oxathiane**	1	3.16	3.17	3.61	0.39
38	**3-(Methylthio)propyl acetate**	3	765.27	461.42	284.71	0
39	Ethyl laurate	1, 8	0.22	0.18	0.24	0.27
40	Isobutyl acetate	1, 7	0.24	0.18	0.33	0.18
41	Phenethyl acetate	1, 2	0.26	0.19	0.23	0.58
42	Hexyl butyrate	1	0.37	0.37	0.39	0.37
43	Isoamyl caprylate	1, 5, 8	0.17	0.11	0.18	0.19
44	Methyl salicylate	3	0.55	0.41	0.49	0.64

Bold indicates that the concentration of the compound exceeds the threshold.

## Data Availability

Data is contained within the article or supplementary material.

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
