# Peer review of "Aromatic Characteristics of Passion Fruit Wines Measured by E-Nose, GC-Quadrupole MS, GC-Orbitrap-MS and Sensory Evaluation"

_foods, 2022, doi:10.3390/foods11233789_

Round 1

Reviewer 1 Report

This paper provides an interesting study in passion fruit wines’ aromatic characteristics using several different approaches. Four different strains of yeasts used in the wine making process were studied. Detailed compound concentrations, as well as sensory evaluations, were reported in this paper.  The correlation reported in 3.7 is also helpful to correlate machine detected compounds to human sensors. The only issue I found in this paper is that the paper stands alone in the field, almost no comparisons or versifications with existing researches. Therefore, here are some suggestions to the paper:

1.       Introduction should include more passion fruit wines researches, especially the authors claim no studies have been conducted to investigate the volatile composition and aromatic features of passion fruit wines. There must be something similar but distinguish the research paper with others.

2.       Four commercial yeast strains must have their specialties. How do those specialties maps with the finding you discovered in this paper reported in the conclusion is another key to link the research with the fermentation field.

3.       Does ranking, which evaluates the likeness of the panels, involved in the sensory evaluations?

4.       Table 2 should be divided into smaller tables using existing type of compounds and put them close to the referenced paragraphs. Current table is way too big to look for relevant data mentioned in the paragraph and decreased the readibility.          

Reviewer 2 Report

The paper "Aromatic characteristics of passion fruit wines measured by 4 different methods" presents an interesting method with good potential in evaluation of food quality. The results are convincing, but I cannot recommend to accept this article in current version unless the authors address the following concerns:

 This paper does not explain the contribution and motivation of this study. Why is this case critical? What's the new contribution?

The paper organization should be explained in the last paragraph in section 1

Before discussing the electronic nose in the last paragraph of the introduction, it is better to discuss the conventional analytical methods and state their problems, and then discuss the electronic nose.

Author should add the novelty of this work in the first paragraph of the introduction section and add a clear hyphothesis in the last paragraph.

The description of PCA in line 144-150 should be removed and moved to part 2-7 Statistical Analysis.

Table S1 should be added to the text of the article because it contains important information.

Line 262 to 266 should be moved to the materials and methods section.

In figure 1b (loading diagram), please draw the circles of this diagram so that the relative role of the sensors can be better defined.

It is better to write the discussion section separately from the conclusion and expand it with previous research. Some relative papers [references] regarding VOC sensors and e-nose may enrich the concepts and background of this work as references:

Food Chemistry 2016, 192, 60-66; doi:10.1016/j.chemolab.2016.11.009 ; https://doi.org/10.1016/j.lwt.2022.113667 ; Foods 2022, 11, 1887; Sensors 2020, 20, 2124

 There are many self-citations, perhaps too many. Since the research topic is addressed by different groups, it would be appropriate for the authors to replace some self-citations with citations from others.

Round 2

Reviewer 2 Report

Thanks for addressing my comments and improving the manuscript. The improvements were satisfactory and I am recommending the manuscript to be published in Foods journal.